# Ranging with Frequency Dependent Ultrasound Air Attenuation

**DOI:** 10.3390/s21154963

**Published:** 2021-07-21

**Authors:** Riccardo Carotenuto, Fortunato Pezzimenti, Francesco G. Della Corte, Demetrio Iero, Massimo Merenda

**Affiliations:** 1Department of Information Engineering, Infrastructure and Sustainable Energy (DIIES), Mediterranea University of Reggio Calabria, 89124 Reggio Calabria, Italy; fortunato.pezzimenti@unirc.it (F.P.); demetrio.iero@unirc.it (D.I.); massimo.merenda@unirc.it (M.M.); 2Department of Electrical Engineering and Information Technologies (DIETI), University of Naples Federico II, 80125 Naples, Italy; fg.dellacorte@unina.it; 3HWA SRL, Spin-Off Mediterranea University of Reggio Calabria, 89126 Reggio Calabria, Italy

**Keywords:** ultrasonic ranging, frequency dependent attenuation, ultrasonic signal

## Abstract

Measuring the distance between two points has multiple uses. Position can be geometrically calculated from multiple measurements of the distance between reference points and moving sensors. Distance measurement can be done by measuring the time of flight of an ultrasonic signal traveling from an emitter to receiving sensors. However, this requires close synchronization between the emitter and the sensors. This synchronization is usually done using a radio or optical channel, which requires additional hardware and power to operate. On the other hand, for many applications of great interest, low-cost, small, and lightweight sensors with very small batteries are required. Here, an innovative technique to measure the distance between emitter and receiver by using ultrasonic signals in air is proposed. In fact, the amount of the signal attenuation in air depends on the frequency content of the signal itself. The attenuation level that the signal undergoes at different frequencies provides information on the distance between emitter and receiver without the need for any synchronization between them. A mathematical relationship here proposed allows for estimating the distance between emitter and receiver starting from the measurement of the frequency dependent attenuation along the traveled path. The level of attenuation in the air is measured online along the operation of the proposed technique. The simulations showed that the range accuracy increases with the decrease of the ultrasonic transducer diameter. In particular, with a diameter of 0.5 mm, an error of less than ± 2.7 cm (average value 1.1 cm) is reached along two plane sections of the typical room of the office considered (4 × 4 × 3 m^3^).

## 1. Introduction

Emerging technologies such as home automation, augmented reality, and gesture interfaces rely on the availability of accurate and fast positioning systems [1,2]. Recently, a large variety of indoor positioning systems (IPS) have proved suitable for many applications, being able to provide cost-effective positioning with sufficiently high speed and accuracy [3,4]. Fast and precise IPS can be used for augmented and virtual reality gestural interfaces [5,6], for navigation in closed places [7,8], for the recognition of human posture and medical rehabilitation [9,10] for the monitoring and care of elderly and disabled people [11], etc. Applications so far recognized for IPS include home automation, robotics, safety, accident prevention through the recognition of dangerous postures and positions of workers, logistics, inventory monitoring, monitoring of body and limb position during sports exercises and training military, game console, monitoring of structures [12], and monitoring of assets and security [13,14]. Certainly, in the near future, positioning systems capable of locating an object with adequate spatial and temporal resolution may enable new possible applications.

Typically, the positioning of a mobile unit or sensor is calculated through a two-step process. First, the distances, or ranges, of the mobile unit from some fixed reference points (RP) are measured. In the second step, these distances are used to geometrically determine the position of the mobile in the reference system defined by the fixed RPs. The ranges necessary for the geometric calculation of the sensor position can be obtained with the desired accuracy and with reasonable cost using the ultrasonic signal time-of-flight technique. With this technique, an ultrasonic traveling signal is emitted from an emitter toward a receiver, and the time of flight (TOF) is measured, which is the time elapsed from the time of emission (TOE) from the emitter to the time of arrival (TOA) at the receiver. In order to estimate this time interval as TOF = TOA − TOE, some technical difficulties must be overcome. First, when the calculation is done by the receiver, then it must know the instant of emission. This implies close synchronization between emitter and receiver, which requires additional hardware, for example, a radio frequency (RF) communication channel. Based on radio frequency channels, several techniques have been proposed in the literature [15,16]. A second difficulty consists in detecting the correct time of arrival (TOA) at the receiver of the traveling ultrasonic signal. Cross-correlation is the most widely adopted technique to have an accurate and robust TOA estimate. Cross-correlation measures the similarity of transmitted and received signals as a function of the time displacement of one relative to the other. The relative displacement that produces the maximum value corresponds to the TOA. Thanks to its integral nature, cross-correlation shows a reduced sensitivity to disturbances [17].

The monotone signal is certainly the easiest to generate and the most suitable for powering commercially available narrow-band ultrasonic transducers. However, the ambient noise makes it difficult to detect the cross-correlation peak corresponding to the TOA since the cross-correlation of a monotone signal shows many adjacent peaks of similar amplitude. Among the different available techniques [18,19], one of the most significant performance improvements is achieved by employing the linear chirp since its cross-correlation shows a very sharp and easily recognizable peak [20,21,22].

One of the most commonly used methods to derive the sensor position starting from the emitter-sensor distances is trilateration, or multilateration in the case of more than three distance measurements. Multilateration uses the distances between RP and the point to be located as radii of spheres, at the intersection of which is the position sought. In 3D space, the minimum number of spheres, and therefore of RP, is four, which drops to three if only calculating position in a half-space is required. On the other hand, information from additional distance measurements can be used to refine the estimated sensor position, thus making it less susceptible to measurement errors [23].

Some positioning systems do not require any emitter–receiver synchronization; they do not estimate directly the single distance between each RP and the mobile unit, but they measure the time difference between the arrival times of the signals emitted simultaneously by several emitters, also called time difference of arrival (TDOA) [24,25]. From the estimated time differences, the sensor position is calculated as the intersection of three hyperboloids. However, such a mathematical formulation requires at least four RPs for 3D positioning within a half-space, which is unfavorable compared to the intersection of the spheres which only requires three RPs. Furthermore, the hyperboloid intersection-based solution of the TDOA positioning problem is highly nonlinear and much more sensitive to ranging errors than the intersection of the spheres. Moreover, it is worth noting that it is not possible to find the emitter–receiver distance by using only one emitter–receiver pair without having any kind of synchronization. From what has been described, it therefore can be seen that to obtain a reliable distance measurement it is necessary to use a technique that requires shaped signals and a significant computational resource to calculate their cross-correlation [26]. Inevitably, from the realization point of view, this translates into a sensor equipped with a processor capable of performing the cross-correlation at three or four times the positioning rate, since three or four distances are needed to calculate the positioning. In addition, the sensor must also have an RF section (or equivalent) to handle the synchronization signals.

With the aim of reducing the complexity of the measurement process and of the sensor hardware, an entirely new method is proposed here for obtaining the distance measurement between the emitter and the sensor. In fact, the proposed distance measurement does not use the flight time of ultrasonic signals between emitter and receiver, but the new technique exploits the attenuation profile of the signal traveling in the air [27], which is a function, among others, of the distance between the emission point and the point of reception. By measuring the amount of attenuation suffered by signals emitted at different frequencies, the distance between emitter and receiver is obtained with simple calculations.

The paper is structured as follows. Section 2 presents the ranging method in detail, while the simulation set-up and numerical results are described in Section 3. Section 4 draws the conclusions of the work.

## 2. Ranging Technique Based on the Frequency Dependent Attenuation

The purpose of the proposed technique is to measure the distance between two points in three-dimensional space using an emitter and a receiver of a suitable ultrasonic signal, without any type of synchronization. The acoustic wave that propagates in the air undergoes energy losses due to the molecular frictions that develop in the medium itself, the extent of which depends, in addition to the medium, on the surrounding conditions. However, the attenuation in air depends mainly on relative humidity (RH). In Bass et al. [27], an experimentally obtained absorption curve in air is presented, which relates each RH level and each frequency of the propagating acoustic wave with a value of the absorption or attenuation coefficient.

Consider a sinusoidal signal with pulsation ω, amplitude *A*, and initial phase *β*:(1)s=Asin(ωt+β)

Furthermore, suppose that there is a line-of-sight (LOS) of length *d* between the emitter and receiver, which is a direct path without obstacles. The received signal *r* by the sensor at point *P* (*d*, *θ*, *φ*) (see Figure 1) first undergoes geometric attenuation, which depends point-by-point on the emission diagram of the emitter:(2)r=D(d,ϑ,φ)s=D(d,ϑ,φ)Asin(ωt+β)
where D(d,ϑ,φ) represents the radiation diagram of the emitter including the effect of geometric attenuation. Due to the presence of energy absorption in the propagation medium, an exponential term must be considered in addition [26], included in the following equation:(3)$$r=D\left(d,\vartheta ,\varphi \right)Asin\left(\omega t+\beta \right)e^{-\alpha d}=Rsin\left(\omega t+\beta \right)$$
where $R=D\left(d,\vartheta ,\varphi \right)e^{-\alpha d}$ is the amplitude of the received sinusoidal signal *r* and *α* is the attenuation coefficient, the latter assumed constant throughout the space of interest for all the time necessary for completion of ranging operations. This is an acceptable assumption when considering an air-conditioned home or office without particularly humid or dry areas.

Knowing the signal emitted *s* and the received signal *r* after propagation along a straight path without obstacles of length *d*, the latter can be estimated with the relationship:(4)$$d=\frac{1}{\alpha }\ln\left[\frac{D\left(d,\vartheta ,\varphi \right)s}{r}\right]$$

Knowing the amplitude of the emitted signal *A* and that of the received signal *R*, the estimate of *d* is still obtained:(5)$$d=\frac{1}{\alpha }\ln\left[\frac{D\left(d,\vartheta ,\varphi \right)A}{R}\right]$$

However, for a correct estimate of *d* it is also necessary to know with a sufficient degree of accuracy D(d,ϑ,φ), which, for any given emitter, depends on the position P(d,ϑ,φ) of the receiver. Since D(d,ϑ,φ) makes the received signal amplitude dependent on the position P(d,ϑ,φ) of the receiver, which is unknown, Equation (5) is not applicable in practice. Furthermore, in general, the actual radiation pattern D(d,ϑ,φ) could be unknown or known with insufficient accuracy. For example, it may depend on the arrangement of reflective surfaces in the space region of interest.

Let us now consider two signals emitted simultaneously by the same emitter, for example, two sinusoids of amplitude *A*_1_ and *A*_2_ with two pulsations *ω*_1_ and *ω*_2_, respectively:(6)$$s_{1}=A_{1}sin\left(\omega_{1}t+\beta \right)s_{2}=A_{2}sin\left(\omega_{2}t+\beta \right)$$

The total emitted signal is $s=s_{1}+s_{2}$. It is worth noting that the same reasoning applies more generally to each pair of sinusoids (*h*, *k*), with *h* ≠ *k* and *h*, *k* ∈ {1, 2, … *n*}, formed by choosing them two-by-two from a set of *n* sinusoids with pulsation *ω_h_* and *ω_k_*, respectively. At the receiver, the received signal *r* is suitably filtered selectively in frequency to yield two signals *r*_1_ and *r_2_*, corresponding to the emitted components *s*_1_ and *s*_2_:(7)$$\begin{array}{c} r_{1}=D_{1}\left(d,\vartheta ,\varphi \right)A_{1}sin\left(\omega_{1}t+\beta \right)e^{-\alpha_{1}d}=R_{1}sin\left(\omega_{1}t+\beta \right)\\ r_{2}=D_{2}\left(d,\vartheta ,\varphi \right)A_{2}sin\left(\omega_{2}t+\beta \right)e^{-\alpha_{2}d}=R_{2}sin\left(\omega_{2}t+\beta \right) \end{array}$$

Considering the ratio *Q* of the signals *r*_1_ and *r*_2_ received as *r* at point *P*, we obtain:(8)$$Q\left(d,\vartheta ,\varphi ,A_{1},A_{2},\alpha_{1},\alpha_{2},\omega_{1},\omega_{2}\right)=\frac{r_{1}}{r_{2}}=\frac{D_{1}\left(d,\vartheta ,\varphi \right)A_{1}sin\left(\omega_{1}t+\beta \right)e^{-\alpha_{1}d}}{D_{2}\left(d,\vartheta ,\varphi \right)A_{2}sin\left(\omega_{2}t+\beta \right)e^{-\alpha_{2}d}}$$
where *R*_1_ and *R*_2_ are the amplitudes of the signals extracted from the received signal *r*, with pulsations *ω*_1_ and *ω*_2_, respectively. The alteration of the signal in the propagation channel consisting of emitter, attenuating propagation medium, and receiver (here for simplicity assumed to have unitary gain) is represented generally for *n* signals by the product $D_{i}\left(d,\vartheta ,\varphi \right)e^{-\alpha_{i}d}$, with *i* = 1, 2, … *n*. Considering only the amplitudes of the signals involved, we obtain:(9)$$Q\left(d,\vartheta ,\varphi ,A_{1},A_{2},\alpha_{1},\alpha_{2}\right)=\frac{R_{1}}{R_{2}}=\frac{D_{1}\left(d,\vartheta ,\varphi \right)A_{1}e^{-\alpha_{1}d}}{D_{2}\left(d,\vartheta ,\varphi \right)A_{2}e^{-\alpha_{2}d}}$$

Knowing a priori the ratio between the amplitudes of the two emitted signals *A*_1_/*A*_2_ = *K*, since it depends on the emission system, we obtain:(10)$$Q\left(d,\vartheta ,\varphi ,K,\alpha_{1},\alpha_{2}\right)=K\frac{D_{1}\left(d,\vartheta ,\varphi \right)e^{-\alpha_{1}d}}{D_{2}\left(d,\vartheta ,\varphi \right)e^{-\alpha_{2}d}}$$

If *ω*_1_ and *ω*_2_ are sufficiently close to each other, then *D*_1_ ≅ *D*_2_, from which it follows that:(11)$$Q\left(d,K,\alpha_{1},\alpha_{2}\right)≅K\frac{e^{-\alpha_{1}d}}{e^{-\alpha_{2}d}}$$

Solving Equation (11) for *d*, we obtain:(12)$$d=\frac{1}{\alpha_{2}-\alpha_{1}}\ln\left[\frac{Q\left(d,K,\alpha_{1},\alpha_{1}\right)}{K}\right]=\frac{1}{\alpha_{2}-\alpha_{1}}\ln\left[\frac{R_{1}}{KR_{2}}\right],$$
where *ω*_1_, *ω*_2_, and *K* are known constants.

Equation (12) shows that, under the hypotheses made, the distance *d* between the emitter and the receiver is obtained, known *K*, *ω*_1_, and *ω*_2_, from the ratio of the amplitudes of the two signals *r*_1_ and *r*_2_, obtained after filtering the received signal *r*. In practice, *r*_1_ and *r*_2_ can be calculated with an FFT, which has the same computational cost as a cross-correlation, but they can also be estimated with a simple narrowband frequency filter, one for each frequency, which requires significantly less computational effort. Please note that *d* cannot assume negative values under the assumption that *ω*_1_ > *ω*_2_ and that the attenuation is monotonically increasing over frequency [27], so that *R*_1_ > *R*_2_. Under these hypotheses, the argument of logarithm is strictly positive, and *d*, too.

Equation (12) can be applied to each pair of sinusoidal signals among *n* sinusoidal signals emitted simultaneously or in sequence, or by considering *n* harmonic components of a single signal of arbitrary shape. In practice, *n* absorption coefficients *α**_i_* (*i* = 1, 2 … *n*) can be easily measured for each pulsation of interest *α**_i_* (*i* = 1, 2 … *n*) in a continuous manner by having a fixed auxiliary microphone at a known distance *l* (see Figure 1) placed in the same environment where the system operates, or obtained from data presented, e.g., in Bass et al. [27] having measured the actual RH with a suitable sensor. Note that the calculation can be done at the desired repetition rate, without the limit determined, for example, by the flight time of the signal from the emitter to the receiver. The emission of signals can be continuous over time or for packets of defined duration. In the latter case, by appropriately choosing the length of the signal packet and the repetition frequency, unwanted reflection phenomena typical of closed environments can be mitigated. The approach presented could work, at least in theory, also considering other propagation media, and could be used for underwater ranging, for example. However, this work is focused on indoor positioning in the air.

## 3. Simulation Setup and Numerical Results

This section provides an overview of the operating principle underlying the simulation software and details on simulation configuration and numerical results.

Setup

The realistic acoustic field emitted by a transducer, including diffractive and attenuation effects, was simulated using the academic acoustic simulation tool Field II. It works in the MATLAB^TM^ environment and it is based on the concept of spatial impulse response [28,29,30,31]. The ultrasonic field for both the pulsed and continuous wave cases is obtained through linear systems. In a first step, the emitted ultrasound field at a specific point in space is obtained as a function of time using the spatial impulse response by applying to the transducer an excitation in the form of a Dirac delta function. Subsequently, by convolving the spatial impulse response with the excitation signal, the field generated by an arbitrary excitation is computed. Any kind of excitation can be considered, based on the theory of linear systems. This technique owes its name, i.e., “spatial impulse response”, to the fact that the impulse response is a function of the spatial position, with respect to the transducer, of the point where the calculated acoustic field is computed [32].

Finally, it is worth noting that, to date, Field II is the only available and reliable acoustic simulator that is not based on a finite element modeling (FEM) approach (e.g., ANSYS, COMSOL, etc.). When dealing with spaces hundreds of times more extended than the typical wavelength considered (less than a couple of cm in the band beyond 18 kHz), as in the case in question, the FEM approach is computationally too expensive. In such cases, the number of nodes is enormous and the calculation becomes very extensive. Instead, the approach used by Field II provides that the calculation of the acoustic field is carried out only in the points considered. This makes the simulation for large spaces very efficient and practically feasible.

However, this approach is partially limited. In fact, the software tool used does not model some important effects in the field of indoor range, such as the phenomenon of reflection. Therefore, it is not possible to easily simulate the reflection of the signal, for example, by acoustically reflective walls, and the phenomena caused by multipath propagation, such as self-interference, typical of even partially reflective environments. Furthermore, the simulator assumes that propagation occurs in free space without considering any obstacles and near-line-of-sight situations. For these reasons, as explained, the simulation results described below are obtained by considering an available line-of-sight between emitter and receiver, and an environment without reflecting walls.

The transducer is represented as follows. The entire surface of the transducer is divided into small rectangles, allowing a transducer surface and field approximations much smaller than the size of the initial element; the smaller the rectangles’ size, the lower the field approximation error. In fact, the distance to the field point is large compared to the size of the rectangles. In general, the element size should be much smaller than the wavelength of the signals used. The calculation is made considering that the rectangular elements behave as if they were rectangular pistons, and knowing the exact impulse response of each [32]. The impulse responses produced by each element at each desired field point is the result of the emission of a spherical wave by each of the small elements [33]. The simulation includes diffractive acoustic phenomena, and the tool gives the possibility to modify the shape and dimensions of the transducer, the signal emitted and to test any ranging or positioning technique intended for application.

The effectiveness of the ranging technique proposed here is evaluated in a typical 4 × 4 × 3 m^3^ room [34]. The simulation results are computed on a grid of points belonging to a vertical section A and a horizontal section B at an height of 1.5 m from the floor (see plane Sections A and B of the room volume, Figure 2). The grid pitch is 5 cm in all directions. In Figure 2, the boundary lines simply represent the extension of the room; however, walls, ceiling, and floor are not considered, since the simulation tool works as if the emission were in free space. The simulated setup has a disc transducer positioned in the center of the ceiling, in position x = 0, y = 0, and z = 0, with the emitting side facing the floor of the room. The transducer central frequency is 20 kHz and it is immersed in air at a temperature of 20 °C, a pressure of 1 atm, and a relative humidity of 55%. Air absorption coefficients of 0.416 dB/m @ 18 kHz and of 0.578 dB/m @ 22 kHz are assumed for the simulation [27]. These values are purely exemplary, since in a real room they may vary from moment-to-moment due to the variation, for example, of the RH. Indeed, to cope with this variability, the proposed system measures the value of *ω*_1_ and *ω*_2_, online during its operation via, for example, the auxiliary microphone (see Figure 1). Moreover, it should be noted that in a real environment three or more digits for the attenuation coefficients are not warranted and were used here for demonstration purposes only. Finally, it is assumed that the actual RH, temperature, and pressure of the real room are sufficiently uniform everywhere.

The shape and size of the emission surface of the transducer determine the emitted and received signals at all points in the space. In this work, circular plane transducers with diameters of 5, 2.5, 1, and 0.5 mm were considered. The circular planar transducers are divided into small square elements with sides 0.025 by 0.025 mm that were used for all the simulations that follow. This element size is a good compromise between the accuracy of the solution and computational resources involved in the simulations.

B.Numerical Results

A summation signal of two sinusoids at *f*_1_ = 18 kHz and *f*_2_ = 22 kHz and duration 10 ms was used as emitted signal for the simulations. The simulation was carried out sampling the signal with a sampling frequency *f_S_* = 10 MHz, to ensure accurate results. In a first step, the numerical simulation computes the acoustic pressure over time generated by the superimposition of the two excitation signals, for each point of the space considered. Subsequently, an ideal receiver is assumed that linearly transduces the pressure signal into an electrical signal, which is then suitably down sampled to 100 kHz and quantized numerically, to simulate a sampling process that is feasible in a real-world device. Finally, the signal amplitudes at each point and the related ranges are calculated through Equation (12). The coefficients *α*_1_ and *α*_2_ that appear in Equation (12) are calculated starting from the attenuation experienced at point *P* by the two harmonic components of the signal. The amplitudes *A*_1_ and *A*_2_ of the two components of the emitted signal were set equal using a value of 1. In this first analysis, uniform white noise was added to the signals received with a reference level of SNR 20 dB calculated at 1 m from the transducer on its emission axis (see point *P* in Figure 2).

The simulation results are shown in the following figures. Figure 3 shows the ranging error committed by using Equation (12) along the vertical section A for four decreasing transducer diameters: 5, 2.5, 1, and 0.5 mm. Figure 4 shows the ranging errors along the horizontal section B for the same transducer diameters. Note the decreasing value ranges reported by the color bars of each subplot when the transducer diameter decreases. Figure 5 shows the cumulative distribution function CDF, i.e., the percent of readings with error less than the value of a given abscissa, for the ranging error along the vertical Section A (blue solid line) and the horizontal Section B (red solid line), respectively. Table 1 and Table 2 summarize the ranging results of the four transducer diameters, reporting mean and maximum ranging errors along the vertical and the horizontal sections, respectively. In Table 1 it possible to appreciate the fast decrease of the mean and maximum error from 602.3 and 1919.2 mm down to 12.5 and 27.2 mm, respectively, when the transducer diameter decreases from 5.0 down to 0.5 mm. In Table 2, with the smallest diameter, the mean and maximum errors reach 11.2 and 27.5 mm, respectively.

C.Discussion

The results obtained clearly show that the proposed numerical method can provide an estimate of the emitter–receiver distance without using the flight time, since the calculation of the ranging through Equation (12) considers only the relative amplitude of the attenuation. The simulation was performed only for two frequencies. By simultaneously using several sinusoids at different frequencies or a broadband signal, it is theoretically possible to obtain a better result as an average of several measurements. The decrease in the ranging error with the decrease in the diameter of the transducer is in agreement with the hypotheses made. In fact, in deriving Equation (12), it was assumed that for frequencies sufficiently close to each other it results $D_{1}≅D_{2}$, and this is especially true when the emitter is reduced in diameter and approaches the isotropic point-like emitter. In fact, by decreasing the diameter of the transducer, the spatial radiation pattern widens, becoming increasingly smooth and similar for the two pulsations *ω*_1_ and *ω*_2_.

In contrast, as shown in Figure 3 and Figure 4, as one goes into the peripheral regions of the acoustic field farthest from the emitter axis, the error increases since *D*_1_ and *D*_2_ differ increasingly. In fact, the region where the error is minimized is the one where *D*_1_ is most similar to *D*_2_, mainly around the axial region. This region, whose three-dimensional shape resembles a cone, widens in space, decreasing the diameter of the transducer.

Certainly, the proposed method does not reach the level of ranging accuracy of many proposed methods that use synchronization, but there are applications that will benefit from the peculiar characteristics of this method, such as personal navigation in malls, airports etc. Even with the accuracy limits discussed, the method still seems to be sufficiently valid for a multiplicity of uses where a not too high accuracy is required, and when the peculiar characteristics of the proposed method take on greater importance: (1) absence of synchronization, which allows the use of a sensor HW of reduced dimensions, since it does not have the RF section, and with less energy consumption compared to sensors that use TOF-based techniques; (2) no limits on the distance measurement rate, since the emitter can emit its signal to the sensor continuously, or with very frequent cycles—from this point of view, the ranging rate is limited only by the onboard computing power; (3) no limitation is imposed by this system architecture on the number of sensors that can coexist in the region of space insonified by the same emitter; (4) the computation of Equation (12) is much less onerous than the computation of a cross-correlation, used by the best ranging techniques based on TOF. On these bases, a wide use of this technique is easily imaginable on mobile devices such as smartphones, tablets, or even notebooks.

## 4. Conclusions

In this work, a new technique was presented to measure the distance between an emitter and a receiver, which is not based on the time of flight, but is instead based on the different attenuation levels that ultrasonic signals of different frequencies undergo when propagating in the air.

The mathematical derivation of the technique was presented together with the validation of the hypotheses through the use of the Field II acoustic simulator. Simulations were conducted assuming free space propagation, and with room temperature 20 °C, relative humidity 55%, and atmospheric pressure 1 atm. The ranging error was calculated along two sections of a typical 4 *×* 4 *×* 3 m^3^ room, one vertical and the other horizontal, at an altitude of 1.5 m from the ground. The performance variation of the proposed technique as a function of the diameter of the emitter was shown. Simulation results show that, using a small diameter emitter aperture, 0.5 mm, and with sufficiently isotropic emission, a ranging error less than ± 2.75 cm and a mean error 1.25 cm were achieved along the two room sections considered.

Subsequently, the merits and limitations of the technique were discussed. The technique works in the absence of synchronization, without intrinsic limits on the distance measurement rate, and with an unlimited number of sensors using the same emitter. However, it does not reach, in its first implementation, the level of accuracy of other measurement techniques based on, for example, cross-correlation. In contrast, this allows for the design of sensors with reduced computational power and thus with reduced dimensions, since they do not require RF sections, and with less computational resources and energy consumption than sensors that use correlation-based techniques. Above all, the fact that it does not require synchronization between emitter and receiver makes this technique imaginable on mobile devices such as smartphones, tablets, or even notebooks, and embedded in chips for IoT or RFID.

## Figures and Tables

**Figure 1 sensors-21-04963-f001:**
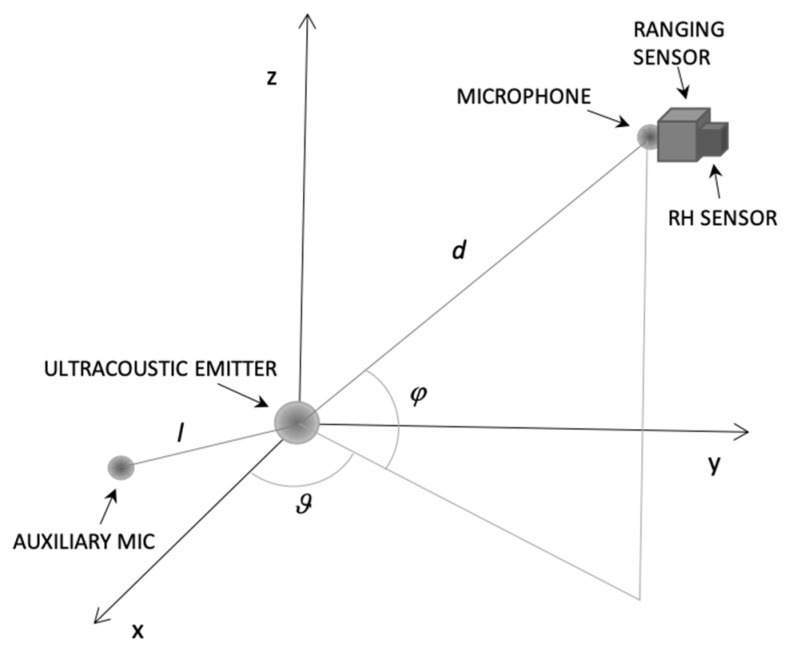
The ranging system: the ranging sensor at point *P* (*d*, *θ*, *φ*), equipped with a microphone and some processing resources, measures the distance *d* from the emitter thanks to the frequency dependent attenuation of the air. The attenuation coefficients are measured online along the known distance *l* using the auxiliary microphone.

**Figure 2 sensors-21-04963-f002:**
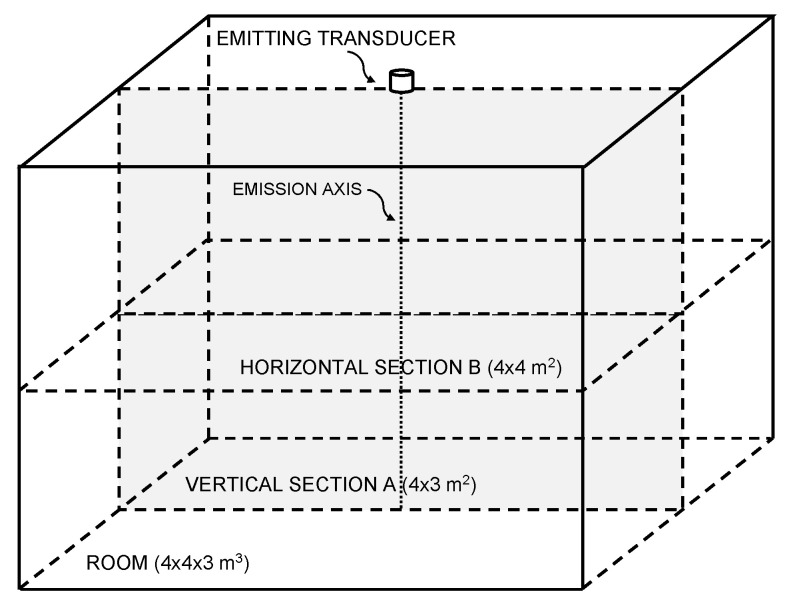
Simulation setup: horizontal and vertical plane sections of the typical 4 × 4 × 3 m^3^ room along which the ranging calculations using cross-correlation are computed. The SNR is considered at point *P*, at distance 1 m from the emitter surface center and on its emission axis.

**Figure 3 sensors-21-04963-f003:**
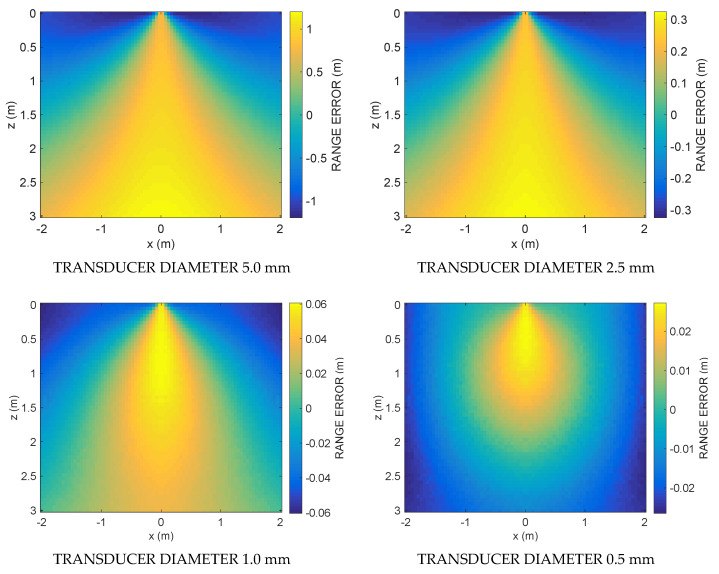
Simulation results: the first column shows the ranging error along the vertical section A for four decreasing values (5.0, 2.5, 1.0, 0.5 mm) of the transducer diameter. Note the decreasing value ranges reported by the color bars when the transducer diameter decreases.

**Figure 4 sensors-21-04963-f004:**
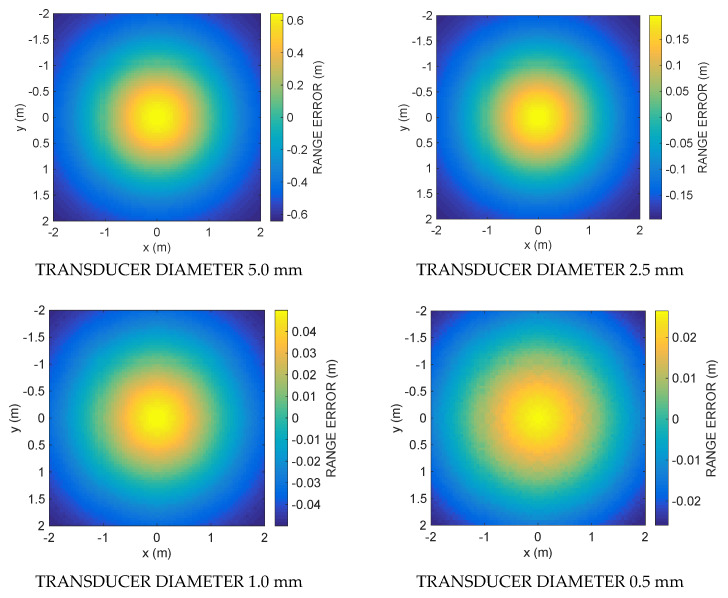
Simulation results: the ranging error along the horizontal section B for four decreasing values (5.0, 2.5, 1.0, 0.5 mm) of the transducer diameter. Note the decreasing value ranges reported by the color bars when the transducer diameter decreases.

**Figure 5 sensors-21-04963-f005:**
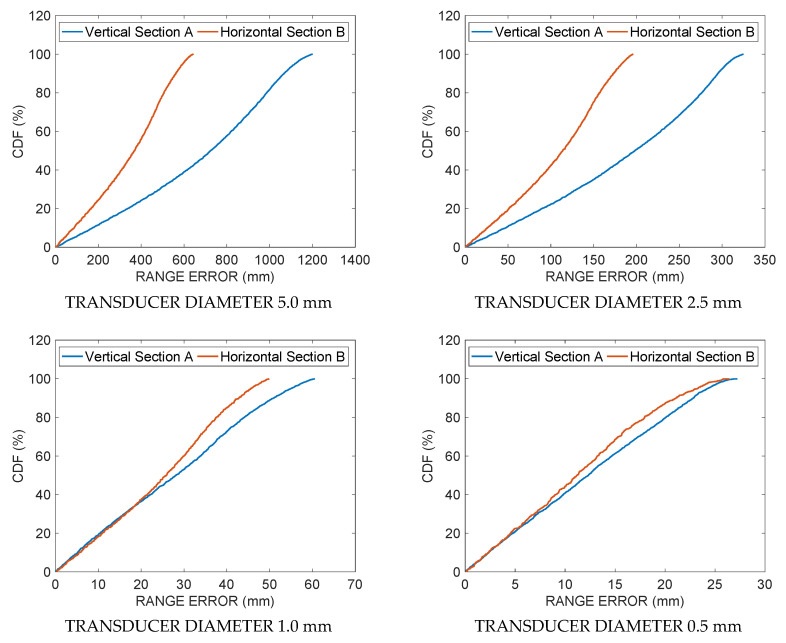
Simulation results: cumulative distribution functions CDF, i.e., the percent of readings with error less than the value of a given abscissa, for the ranging error along the vertical Section A (blue solid line) and horizontal Section B (red solid line) for four decreasing values (5.0, 2.5, 1.0, 0.5 mm) of the transducer diameter.

**Table 1 sensors-21-04963-t001:** Mean and maximum ranging error as a function of the transducer diameter along the vertical Section A.

Transducer Diameter (mm)	Range Absolute Mean Error (mm)	Range Absolute Maximum Error (mm)
5.0	602.3	1919.2
2.5	171.5	524.4
1.0	30.8	89.9
0.5	12.5	27.2

**Table 2 sensors-21-04963-t002:** Mean and maximum ranging error as a function of the transducer diameter along the horizontal Section B.

Transducer Diameter (mm)	Range Absolute Mean Error (mm)	Range Absolute Maximum Error (mm)
5.0	603.7	1015.7
2.5	175.7	297.2
1.0	32.4	63.4
0.5	11.2	27.5

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
