# Peer review of "Ranging with Frequency Dependent Ultrasound Air Attenuation"

_sensors, 2021, doi:10.3390/s21154963_

Round 1

Reviewer 1 Report

The ranging approach proposed is based on the atmospheric absorption of sound which is frequency-dependent. By emitting a mixed signal with at least 2 pure frequencies, the authors argue that a receiver can self-calculate its relative distance from the source.

In my reading of the paper there are several aspects that appear incomplete and therefore I cannot recommend its publication. 

Equation (12) is the author's main result, however, it appears to have a few problems. Firstly, it can return a negative distance, especially in the presence of frequency-selective additive noise. Second, it assumes that the two emitted frequencies are similar enough (i.e., w1 \approx w2 ) implying that the absorption coefficients are also similar (i.e, a1 \approx a2) resulting in dividing by a small number in equation 12 making it highly unstable. Both these aspects are quite important limitations of the proposed method that should be explored in more detail.

The paper is then concerned with the emitting transducer diameter, which the authors simulate from 5mm to 0.5mm, but do not consider the effect it has on the directivity pattern nor on the frequency response of the transducer. Is that not important?

Figures 3 and 4 plot the ranging error which seems to be minimized along some diagonal/cone region. The authors do not say why this is so, nor why this depends so strongly on the transducer size. 

Finally, the authors claim that a major benefit of their ranging method is that the computation of (12) is much less onerous than the computation of a cross-correlation, however, this assumes that one already has performed an FFT to obtain r1 and r2, which is essentially the same computation the same as a cross-correlation computation.

The paper can be improved significantly along the few points that I have raised.

However, to be taken seriously as a potential novel ranging and positioning candidate then one would really have to look at demonstrating an experimental proof of concept system (which I understand is beyond the scope of this paper).

Reviewer 2 Report

An innovative method is proposed for measuring the distance between the transmitter and receiver using 20 ultrasonic signals in the air. The idea is described clearly. But the authors can improve their work in the following parts:

-The introduction can be improved by regrouping the chapter and using subchapters. (e.g. IPS, .... distance measurments methods (TDOA..) , detections methods..). Currently there are too many sections and different topics.

-Chapter 2 "Ranging Technique based on the Frequency Dependent" can also be improved. The outline of the chapter can be improved with fewer sections (e.g. lines 226- 244 - currently each sentence has a new section).

-Please enlarge the captions of the illustrations.  There is still space available. 

-The main problem of the paper is the correct validation of the simulation. . The described idea has potential. Showed simulation results are important, but these should be compared with practical measurements or at least further validation.

Round 2

Reviewer 1 Report

Thank you for addressing my previous concerns.

My only last concern is that the paper does not address any ranging literature and results from the underwater ranging community (something that I am not familiar with myself).

Reviewer 2 Report

The paper can be still improved:

-There are still too many sections. I suggest to reduce the sections further. It is then easier to read.  One sentence (or two or three line) paragraphs should be avoided. E.g. 53-58, 100-110, 280-283

Round 3

Reviewer 2 Report

I suggest to reduce the sections further. Lines: 123-132
